# The Challenge of Diagnosing Labyrinthine Stroke—A Critical Review

**DOI:** 10.3390/brainsci15070725

**Published:** 2025-07-07

**Authors:** Alexander A. Tarnutzer, Sun-Uk Lee, Ji-Soo Kim, Diego Kaski

**Affiliations:** 1Neurology, Cantonal Hospital of Baden, 5404 Baden, Switzerland; 2Faculty of Medicine, University of Zurich, 8006 Zurich, Switzerland; 3Neurotology and Neuro-Ophthalmology Laboratory, Korea University Medical Center, Seoul 02841, Republic of Korea; sulee716@gmail.com; 4Department of Neurology, Korea University Medical Center, Seoul 02841, Republic of Korea; 5Dizziness Center, Clinical Neuroscience Center, Seoul National University Bundang Hospital, Seongnam 13620, Republic of Korea; jisookim@snu.ac.kr; 6Department of Neurology, Seoul National University College of Medicine, Seoul 13620, Republic of Korea; 7SENSE Research Unit, Department of Clinical and Movement Neurosciences, Institute of Neurology, University College London, 33 Queen Square, London WC1N 3BG, UK; d.kaski@ucl.ac.uk

**Keywords:** acute vestibular syndrome, vertigo, hearing loss, stroke, inner ear, imaging

## Abstract

Acute vertigo or dizziness that is accompanied by a sudden sensorineural hearing loss (SSNHL) often poses a diagnostic challenge. While a combined audiovestibular deficit makes an inner ear pathology most likely, this does not necessarily exclude a vascular pathology that may be a harbinger of future sinister events. This is especially true for strokes within the territory of the anterior inferior cerebellar artery (AICA), because the labyrinth receives its vascular supply most often by branches of the AICA. Thus, acute labyrinthine ischemia may present in combination with focal neurologic deficits, but also in isolation or as a warning sign before focal stroke signs arise. How can labyrinthine ischemia be differentiated from an idiopathic SSNHL? In this critical review, we discuss both the pathophysiology and the differential diagnosis of acute audiovestibular deficits. We will also address the value of state-of-the-art MR imaging in visualizing labyrinthine ischemia. Finally, we will discuss treatment options and review the prognosis of acute audiovestibular deficits.

## 1. Introduction

Acute inner ear impairment may present with either vestibular (e.g., vertigo, dizziness, gait imbalance) or cochlear symptoms (e.g., hearing loss, aural fullness, tinnitus) or a combination of both. The broad range of underlying causes, often lacking obvious focal neurologic signs and negative brain imaging results, poses significant challenges for the clinicians in charge [1]. At the same time, acute vertigo and dizziness are amongst the most frequent presenting symptoms in the emergency room (ER) environment, constituting between 2.1% and 4.4% of all admissions [2,3,4]. This translates to approximately 4.4 million consultations per year in the United States and probably to 50 to 100 million consultations worldwide [5].

Amongst all acutely dizzy patients, about 3–5% will be eventually diagnosed with an ischemic stroke [5], in line with the diagnostic criteria for vascular vertigo and dizziness that were published by the Bárány Society [6]. Thus, a critical aim in the ER setting is to correctly identify and treat this subset of patients. If vertigo or dizziness is acute and persistent (i.e., lasting for more than 24 h) and is accompanied by motion intolerance, nausea/vomiting, gait imbalance, and often also nystagmus, this is referred to as acute vestibular syndrome (AVS) [7]. Approximately 25% of all AVS cases turn out to suffer from a central (mostly ischemic stroke) cause [8]. Amongst all vertebrobasilar strokes presenting as AVS, those involving the territory of the anterior inferior cerebellar artery (AICA) are the most difficult to diagnose. This is mainly because focal neurologic signs that would support a central origin may be missing in up to two-thirds of all central AVS cases [9]. Furthermore, ischemic strokes in the AICA territory may involve both central and peripheral vestibular structures, as the vascular supply of the inner ear originates most often from branches of the AICA. Even using refined oculomotor bedside tests such as the HINTS battery (i.e., head-impulse, nystagmus, and test of skew [10]), the sensitivity for detecting a central lesion is only 84.0% ([95% CI = 65.3–93.6]) if the AICA territory is involved [9]. Thus, about one out of seven cases is misclassified as peripheral (i.e., acute unilateral vestibulopathy or acute labyrinthitis) in this subset of patients. This misclassification is mostly due to the interrupted angular vestibulo-ocular reflex (aVOR) arc, where the resulting unilaterally abnormal head-impulse test is interpreted as “peripheral” in origin. When taking into account hearing loss as well (as implemented by the HINTS plus (+) algorithm [11]), the sensitivity for detecting a central cause at the bedside can be increased substantially to 95.7% [95% confidence interval (CI) = 79.0–99.2] [9]. However, ischemic stroke in the AICA territory may also be restricted to the labyrinth, causing an acute cochlear and/or vestibular loss of function without any focal neurologic signs and no central lesions on brain MRIs, including diffusion-weighted imaging (DWI).

A key differential diagnosis in unilateral acute hearing loss is sudden sensorineural hearing loss (SSNHL), which may be unilateral (rarely bilateral), occurring within a 72 h window, and is defined as a decrease in hearing thresholds of ≥30 dB affecting at least three consecutive frequencies [12]. SSNHL affects 5 to 27 per 100,000 people annually, with about 66,000 new cases per year in the US. In a meta-analysis of 23 publications, a vascular cause was identified in 3% of cases, whereas in 71 to 85% of all SSNHL patients, no underlying cause could be identified; thus, they were classified as idiopathic SSNHL [13]. However, this classification was based on a negative MRI scan (for stroke), and yet most AICA vascular events causing hearing loss are known to be too small to be seen on routine imaging, particularly outside the critical imaging window (24–72 h after symptom onset). Furthermore, 8% of all vertebrobasilar strokes were accompanied by SSNHL in a one case series, with 79% of cases being located in the AICA territory [14]. Thus, identifying those patients who suffer from an ischemic labyrinthine stroke is an ongoing challenge.

In this critical review, we will focus on the diagnostic workup and management of the patient with acute audiovestibular symptoms. In order to minimize selection bias of the literature discussed here, we have performed a systematic literature review through MEDLINE using a search string and predefined exclusion criteria. We will summarize the findings from those studies included, reporting both on the clinical and imaging features identified, and place those in relation to the inner ear anatomy and vascular supply. In addition, we will discuss recent developments in inner ear imaging that allow for a more reliable distinction between labyrinthine inflammation, hemorrhage, and ischemia, and how MR imaging can be combined with bedside examination to maximize the diagnostic yield.

## 2. Inner Ear Anatomy and Vascular Supply

The vascular supply of the inner ear is provided by the labyrinthine artery, which is a single terminal artery with minimal collaterals that originates most often (83.6%) from the AICA and occasionally (in 12.3%) from the basilar artery [15,16] (see Figure 1). It consists of three branches as it enters the inner ear: the anterior vestibular artery (AVA), the main cochlear artery (MCA), and the vestibulocochlear artery (VCA), with the latter two originating from the common cochlear artery [17]. The AVA irrigates the anterior and lateral semicircular canals, the utricle, and a small part of the saccule. The MCA is in charge of the blood supply to the apical three-fourths of the cochlea, i.e., its apical and middle turns. The VCA supplies the basal one-fourth of the cochlea through the cochlear branch of the VCA, and the saccule and the posterior semicircular canal through the vestibular branch of the VCA (i.e., the posterior vestibular artery). The arterial supply to the cochlea can be MCA-dominant or VCA-dominant [15].

For the central processing and forwarding of peripheral–vestibular inputs, the vestibular nuclei are the key structure. They are located in the caudal dorsal paramedian pons at the pontomedullary junction [19,20]. Thus, lesions involving the vestibular nuclei or the dorsal root entry zone of the eighth cranial nerve may result in a reduced response during caloric irrigation, head-impulse tests, or measurements of the dynamic visual acuity.

Cochlear inputs are forwarded through the cochlear branches of the vestibulocochlear nerve to the cochlear nuclei, which are located in the dorsolateral medulla. The central auditory pathways then cross and ascend through the lateral lemniscus, reaching the inferior colliculus, the medial geniculate nucleus, and eventually the primary auditory cortex [21]. Unilateral acute central hearing loss may occur with a lesion at the root entry zone of the eighth nerve (pontomedullary junction), at the cochlear nucleus level (dorsolateral medulla) [22,23], or at the level of the contralateral pons [24].

## 3. Methodology and Results of the Literature Review Performed

We searched MEDLINE through PUBMED for English language articles. The search strategy was designed by a clinical investigator with relevant domain expertise in neurology (AAT). We relied on the following strategy and looked for the listed specific components in all articles: (1) acute/new onset of symptoms, (2) clinical or radiologic signs of ischemic/hemorrhagic stroke, and (3) involvement of the labyrinth and/or the anterior inferior cerebellar artery (AICA). We then selected a series of textual terms to enter into the search system that would refer to the selected criteria (see the Appendix A for details on the search strategy and PRISMA flow chart [25]). A manual search of the references of eligible articles was also performed. We did not seek to identify research abstracts from meeting proceedings or unpublished studies. Since the submitted work is a critical review, ethical approval was not necessary.

When reviewing all of the identified citations for eligibility, we used predetermined exclusion criteria and a controlled methodology to select the relevant studies (see the Appendix A for details on the search strategy). The review was conducted by a single rater (AAT). Only English language articles with original data on human subjects with a labyrinthine or AICA stroke, which reported on vertigo, dizziness, gait ataxia, and/or hearing loss, were included.

## 4. Results from the Literature Review

Our search was performed on 19 May 2025 and identified 634 unique citations. Of the 634 papers screened, 557 (87.9%) were excluded at the abstract level. We further examined 88 manuscripts at the full-text level, including 11 additional manuscripts after reviewing the citations of selected manuscripts. While 75 were considered eligible, 13 were excluded for the following reasons: four did not report on patients with AICA or labyrinthine involvement, four did not address patients with stroke or hemorrhage, three did not contain data on human subjects with labyrinthine/AICA stroke, and two did not report on vertigo, dizziness, gait imbalance, or hearing loss. Furthermore, 3 out of those 75 studies [26,27,28] reported preliminary findings of one study [14] and were, thus, not further considered. Importantly, there is a potential further overlap among single studies; thus, we cannot exclude that some of the labyrinthine strokes identified in our review were counted twice [14,29,30].

In total, we identified 3011 patients with SSNHL and/or acute vertigo/dizziness from 72 studies (see Table 1 for details). The study sample size ranged from 1 to 1300 patients, with most large studies focusing on an SSNHL of mixed etiology, pointing to substantial heterogeneity amongst included studies (this was most likely related to the retrospective nature of most of the studies included and the varying aims and inclusion criteria). Noteworthy, however, is that none of the studies reporting on strokes in SARS-CoV-2 patients contained sufficient details on the stroke location in order to be included in our review.

A majority of patients were included based on their cochlear symptoms (i.e., SSNHL) (89.3%), whereas a minority was included based on a combined audiovestibular deficit (4.0%) or vestibular symptoms (i.e., AVS) only (2.8%). Amongst all included patients, 45.9% reported dizziness or vertigo, whereas 96.1% of patients suffered from SSNHL.

Focal neurologic symptoms were present in 5.0% and absent in 79.1% of patients. Information was lacking in 15.9% of patients. Information on the presence/absence of the subtle oculomotor findings (as seen on assessing a patient’s gaze stability in primary gaze [looking for spontaneous nystagmus], eccentric gaze [looking for gaze-evoked nystagmus], and during positional testing, assessing their vertical gaze stability [looking for vertical divergence on an alternating cover test], and the integrity of the aVOR [when performing the head-impulse test]) was available only in 34.0% of cases, with confirmed signs in 5.4% of patients only.

Amongst those patients that received MR imaging (93.9%), a 3D-FLAIR (fluid-attenuated inversion recovery) sequence was included in 58.5%, whereas a regular 2D-FLAIR was retrieved only in 30.1% in addition to T1 imaging and DWI. A delayed 3D-FLAIR sequence obtained 4 h after application of an intravenous contrast agent was obtained in 1.9% of all patients.

Peripheral audiovestibular pathologies were reported in 2691 out of 3011 patients (89.3%). The most frequent peripheral diagnoses identified were labyrinthine hemorrhage (16.2%), inflammatory inner ear disorders (3.9%), and tumors (1.4%). Other peripheral diagnoses (20.5% in total) included undetermined, MRI-based descriptive diagnoses of a labyrinthine signal change (high inner ear protein vs. labyrinthine hemorrhage vs. labyrinthine ischemia vs. labyrinthine inflammation), perilymph or labyrinthine fistula, decompression sickness, and neurovascular conflicts (see Table 1 for details).

Central vestibulocochlear pathologies were identified in 323 patients (10.7%). An MRI-DWI positive stroke was found in 235 patients, with 77.9% of strokes involving the AICA territory (being restricted to the AICA territory in 52.3%). Other central causes (n = 88) included various vascular lesions, demyelinating brainstem and cerebellar lesions, and cerebral venous thromboses (see Table 1 for details). In 14.5% of patients (out of all patients having ischemic lesions), the central lesion involved the areas that belong to the central vestibular or auditory pathways and could explain the cochlear and/or vestibular symptoms, whereas this was not the case in the remaining 85.5% of cases. Thus, it is more likely that, in addition to this, an occlusion of the labyrinthine artery (or one/several of its branches) also occurred. However, with the MRI protocol used in these 235 cases, detection of labyrinthine ischemia was not possible, and, therefore, this hypothesis could not be validated. We identified a total of 46 patients (from six studies) with prodromal audiovestibular symptoms that subsequently developed focal neurologic signs and had an AICA stroke confirmed on MRI-DWI.

Vascular peripheral cochleovestibular pathologies were identified in a total of 13 patients. This included five cases with MRI-confirmed labyrinthine ischemia (on 3D-FLAIR sequences obtained 4 h post contrast application in four cases [49,74,94,95] and on 3D-VISTA (volume isotropic turbo spin-echo acquisition) sequences post contrast application in one case [98]) and MRI-DWI positive lesions of the vestibulocochlear nerve in two cases [66,96]. Other pathologies that were identified on MRI, and that likely resulted in peripheral vestibulocochlear dysfunction, were three cases with cerebral venous thrombosis of the sigmoid and transverse sinuses, potentially leading to hemostasis [55]; one case with fibrosis of the left posterior semicircular canal on 3D-FIESTA (fast imaging employing steady-state acquisition) sequences 12 days after the onset of cochleovestibular symptoms [67]; one case with a fibrotic scar of the vestibulocochlear nerve on a histopathology performed 7 years after an acute unilateral cochleovestibular loss and normal brain MRI [73]; and one case of an AICA aneurysm, resulting in local compression of the vestibulocochlear nerve [82].

Information on recoveries from cochlear and/or vestibular symptoms was available for 680 patients (22.6%), and this ranged from no recovery (32.9%), a partial recovery (58.0%), and full recovery (5.6%) (see Table 1 for details).

## 5. Clinical Features Distinguishing an Inner Ear Pathology from a Central Vertebrobasilar Lesion Location

Both a structured history taking and a targeted neuro-otologic examination are key in all patients presenting with acute vertigo, SSNHL, or a combination of both. By following the TiTrATE approach (timing, triggers, and targeted examination [5]), this offers such a framework to clinically address those patients who present with acute vertigo (and additional hearing loss). Asking for accompanying symptoms, such as hemiparesis, diplopia, or limb ataxia, and searching for focal neurologic signs on clinical examination are essential. In those patients with audiovestibular symptoms and additional focal neurologic signs, a central cause is readily suspected, and brain MRI, including DWI, is ordered. However, acute vestibular and/or cochlear symptoms may present in isolation. Distinguishing peripheral from central causes in these patients is more challenging, but here, diagnostic accuracy requires a focused oculomotor bedside testing, including algorithms such as HINTS (+), to identify more subtle central signs.

The oculomotor features that clearly point to a central cause include oculomotor palsies; internuclear ophthalmoplegia; purely vertical (upbeat/downbeat), purely torsional or combined vertical-torsional spontaneous nystagmus [100]; impaired eccentric gaze holding (i.e., gaze-evoked nystagmus); and a presence of a vertical divergence (i.e., skew deviation) on the alternating cover test [101]. Noteworthy, when assessed quantitatively, small-amplitude skew deviation may also be seen in peripheral AVS cases [102]. Similarly, severe gait and truncal instability, i.e., being unable to stand or sit unassisted, are highly predictive for a central cause, with a specificity of 99.1% [98.0–100.0%] [103].

A unilaterally impaired aVOR, as assessed by the horizontal HIT (head-impulse test), has a sensitivity for detecting a central cause in AVS patients of only 79.9% (95% CI = 72.2–87.5), i.e., one out of five strokes are missed if only the HIT is assessed. For the subset of AICA strokes, the sensitivity of the horizontal HIT is much lower, reaching only 36.0% [95% CI = 20.2–55.5] [9]. Thus, these numbers emphasize the need for caution when linking an abnormal HIT with a peripheral vestibular pathology, and should trigger the use of HINTS.

In the setting of AVS, for patients who demonstrate peripheral HINTS (i.e., a combination of a unilaterally abnormal HIT that demonstrates catch-up saccades and no gaze-evoked nystagmus or skew deviation), the presence of a new-onset unilateral hearing loss on the side of abnormal HIT has been shown to be predictive for a central cause when using the HINTS+ algorithm [9,11]. By adding hearing loss to the HINTS algorithm, it is mainly the AICA strokes with a combined peripheral/central pattern that are classified correctly as having an underlying central disorder. Importantly, symptoms (e.g., ear pain) and signs of inflammation (e.g., otorrhea, redness/bulging of the eardrum, or vesicles/crusts) on otoscopy need to be investigated in these patients, as well, to identify infectious inner ear disease. What is noteworthy, however, is that for the subset of isolated AVS, diagnostic accuracy using HINTS (+) remained very high, with a sensitivity of 96.8% for detecting central causes [95% CI = 93.2–100.0] [9]. For isolated SSNHL, no comparable approach has been evaluated, however.

## 6. The Value of Quantitative Audiovestibular Testing

Clinically identified new-onset unilateral hearing loss should be confirmed by pure tone audiometry (PTA), and in patients with acute vertigo or dizziness, quantitative vestibular testing offers the possibility to assess the integrity of the aVOR and otolith pathways. We recognize, however, that these may not be readily available in many emergency centers.

Taking the vascular supply of the distinct peripheral vestibular sensors (i.e., the semicircular canals and the otolith organs) and the cochlea into account, different patterns can be predicted based on the arterial branch affected. In a single case presenting with acute vertigo, dysarthria, and AICA stroke involving the left lateral pons on MRI-DWI, quantitative vestibular testing demonstrated isolated damage to the superior vestibular labyrinth, while the inferior vestibular labyrinth and the cochlea were preserved [58]. This pattern was consistent with selective occlusion of the anterior vestibular artery that supplies the anterior and lateral canals and the utricle [17]. Such selective impairment of the afferents from one part of the vestibular labyrinth, or a partial involvement of several parts, strongly speaks against a central lesion, e.g., at the root entry zone or the level of the vestibular nuclei [19]. Furthermore, it may indicate decreased vulnerability of the inferior vestibular labyrinth due to better collateral blood supply [73]. Similarly, isolated unilaterally abnormal ocular vestibular-evoked myogenic potentials (oVEMPs) and cervical vestibular-evoked myogenic potentials (cVEMPs) in patients with acute audiovestibular symptoms may point to (transient) labyrinthine ischemia [104].

However, in a cohort of patients presenting with vertigo and SSNHL of various causes and varying symptom durations (ranging from days to years), quantitative audiovestibular testing, including video-HIT of all six semicircular canals, oVEMPs, cVEMPs, and PTA, did not provide an easy separation of ischemic from non-ischemic etiologies [84]. These negative findings may potentially be explained by delayed testing (being performed within 14 days of symptom onset in only 11/27 patients) and the lack of a gold standard for confirming a diagnosis (such as delayed 3D-FLAIR imaging). Additionally, an assessment of brainstem auditory-evoked potentials (BAEP) may be helpful in the distinction between labyrinthine and central lesions [26,58]. However, there is currently insufficient evidence to recommend the introduction of specialist audiovestibular testing for acute audiovestibular loss, with the key exception of formal tests of hearing.

## 7. The Role of MR Imaging in Identifying Vertebrobasilar Lesions

Brain imaging using MRI (including DWI, T1, and 3D-FLAIR sequences) is recommended in all AVS patients who present with central HINTS (+) [6], and also in patients with SSNHL [12], in order to identify the underlying structural causes. Importantly, early (i.e., obtained within the first 24–48 h after symptom onset) MRI-DWI in a patient having a vertebrobasilar stroke may lead to false negatives in up to 20% of the cases, and thus repeated, follow-up imaging may be necessary (with a time window from 72 h to 14 days after symptom onset) [9]. The increased sensitivity for delayed imaging likely reflects an acute ischemia being missed due to its small size rather than a limitation of DWI when identifying early ischemia, and the potential for small labyrinthine ischemia to progress to involve the AICA territory with the proximal migration of a clot. CT imaging has very limited diagnostic value in the acutely dizzy patient: a high-resolution temporal bone CT scan may be useful for identifying inner ear pathologies, such as bone defects in the labyrinthine or perilymph fistula, or bacterial labyrinthitis. Brain CT angiography can help to triage acute revascularization therapies by indicating occlusion of a target vessel (e.g., basilar artery or AICA) and by exclusion of contraindications for acute treatment (such as a hemorrhage or subacute stroke).

The primary aim of MR imaging in patients with acute unilateral cochleovestibular symptoms is to identify the central (mostly ischemic) causes. Thus, previous publications have linked acute cochlear [14], cochleovestibular [30,77], or vestibular symptoms [58] in the presence of MRI-DWI-positive lesions to combined peripheral–central ischemic lesions. Importantly, central auditory and vestibular pathways may be affected by AICA strokes, as well, especially when the dorsal/dorsolateral parts of the pontomedullary junction or the pons are affected (see also the separate section further above). In these cases, the central lesion(s) may explain the audiovestibular symptoms, as well. Such central lesions were identified in 18.6% of all vertebrobasilar strokes, including the AICA territory, in our review. Thus, in those patients with confirmed vertebrobasilar stroke who do not show involvement of the central auditory or vestibular pathways, the presence of SSNHL, unilaterally impaired HIT, and unilateral caloric paresis points to labyrinthine ischemia [98]. Conversely, bilaterally normal responses during caloric irrigation were considered to indicate a central origin of the audiovestibular symptoms in AICA stroke in one study [30]; however, such investigations are rarely available acutely.

## 8. The Value of MR Imaging in Localizing Vascular Pathologies to the Inner Ear

While the presence of a vertebrobasilar stroke and acute unilateral audiovestibular symptoms (with the central auditory and vestibular pathways being spared on MRI-DWI) is indicative of a labyrinthine origin of the patient’s symptoms, imaging of labyrinthine stroke is challenging. The established MRI protocols used in a suspected stroke are not optimized to detect labyrinthine ischemia, which explains the very low rate of reported cases. In our literature review, labyrinthine ischemia was confirmed mostly using advanced MR imaging, including delayed MR imaging 4 h after the application of an intravenous contrast agent [49,74,94,95]. Although rare, a labyrinthine infarction can be documented on a 2D-FLAIR or 3D-FLAIR with gadolinium enhancement [98]. Meanwhile, other peripheral vascular pathologies that result in acute audiovestibular symptoms may be detected on MRI. These include ischemic lesions of the vestibulocochlear nerve (as demonstrated on MRI-DWI in single cases [66,96]), fibrosis of the left posterior semicircular canal on 3D-FIESTA sequences 12 days after the onset of audiovestibular symptoms [67], and an AICA aneurysm, which results in a local compression of the vestibulocochlear nerve [82] in single cases. Peripheral vestibulocochlear dysfunction, in the context of MRI-confirmed cerebral venous thrombosis of the sigmoid and transverse sinuses, may be potentially explained by the hemostasis and hypoperfusion of the labyrinth, as was discussed in a small series of three cases [55].

Thus, in those cases with negative standard MRI, additional delayed post-contrast 3D-FLAIR sequences, imaging of the cerebral venous sinuses, and the search for AICA aneurysms on MR angiography may be considered to increase diagnostic accuracy.

## 9. Labyrinthine Stroke as a Warning Sign of Vertebrobasilar Stroke

Acute isolated unilateral audiovestibular symptoms may be a warning sign for vertebrobasilar stroke [105]. We identified a total of 46 patients (from six studies) with prodromal (transient or persistent) audiovestibular symptoms who subsequently developed focal neurologic signs and had an AICA stroke that was confirmed on MRI-DWI [14,52,62,72,77,97]. Such prodromal symptoms may be observed in a substantial fraction of AICA strokes, with reported numbers of 7.4% [72] and 31% [14] in two case series. Thus, in the setting of isolated new-onset (transient and often lasting a few minutes only [26]) audiovestibular symptoms and negative MRI-DWI imaging (including T1 and 3D-FLAIR sequences to identify other inner ear pathologies and abnormalities in the cerebellopontine angle), suspicion of a labyrinthine stroke should be kept high. Secondary preventive measurements, including antiplatelet therapy, should be initiated, and a stroke workup should be performed in the setting of a monitored stroke unit. One study found that those patients with SSNHL and who had three or more stroke risk factors, a bilateral SSNHL, a moderately severe to total SSNHL, and any intracranial high-grade (i.e., > 50%) large artery stenosis or occlusion, were at a higher risk of developing ischemic stroke during hospitalization [52]. Supporting this observation, a systematic review and meta-analysis found that patients with SSNHL face a higher risk of stroke than those with age-related hearing loss [106]. Thus, timing and symptom evolution may have a critical impact on whether the right diagnosis is made or not in such cases with warning signs. Closely following up on these patients is recommended.

## 10. The Spectrum of Other Causes of SSNHL and Acute Audiovestibular Loss and the Role of Imaging

The most frequent peripheral diagnoses made on MR imaging of the labyrinth were labyrinthine hemorrhage (16.2%), inflammatory inner ear disease (3.9%), and tumors of the cerebellopontine angle (1.4%). For other peripheral causes, such as decompression sickness, many details were lacking, making a validation of this diagnosis difficult [99]. In patients with SSNHL, MR imaging of the inner ear remained negative in 57.7% of cases in our literature review, resulting in a diagnosis of “idiopathic” SSNHL. In an additional 14.4% of cases, MR imaging of the labyrinth was abnormal, but no distinction was possible between inflammatory changes (“high protein”) and labyrinthine hemorrhage.

One potential explanation for the high rate of undetermined cases is the MRI protocol used in these studies. Importantly, even when using 3D-FLAIR sequences (see [107] for a review), a distinction between a minor labyrinthine hemorrhage and high protein was not reliably possible in several publications [51,53,54,93], which could also be related to a lack of dedicated internal auditory canal sequences. Thus, others suggested the use of dedicated 3D-FLAIR VISTA sequences in these patients [42]. Furthermore, early MRI, including 3D-FLAIR sequences, may provide false negatives in cases of labyrinthine hemorrhage because of a lack of methemoglobin and protein accumulation in the hyperacute setting [41,71].

In contrast, imaging of the cerebellopontine angle, including contrast-enhanced sequences, will reliably detect tumors affecting the eighth cranial nerve (most often the vestibular schwannoma and sometimes also the meningioma), whereas intrameatal or intralabyrinthine schwannoma may be missed on routine MR imaging [108].

## 11. Treatment Options and the Prognosis of Labyrinthine Stroke

In patients with suspected or confirmed labyrinthine stroke, either in combination with vertebrobasilar stroke on MRI-DWI or in isolation, a diagnostic workup in the setting of a monitored stroke unit is recommended. Acute revascularization treatment options can be considered in limited cases, particularly when a labyrinthine infarction is highly suspected based on the clinical characteristics or neurotologic evaluation and with a confirmed vessel occlusion in the vertebrobasilar territory (especially of the AICA), or among individuals whose hearing or balance is an occupational necessity (e.g., musicians or artists). However, intravenous thrombolysis or endovascular treatment in patients presenting with a central AVS is rarely done, as recently discussed in a systematic review [109]. Furthermore, we are unaware of any published cases that treated isolated labyrinthine ischemia with intravenous thrombolysis. For secondary prevention, antiplatelet therapy (or anticoagulation if indicated) should be initiated, and the underlying cause of the stroke should be investigated using the established guidelines [110]. Furthermore, what is likely is that a substantial fraction of (isolated) labyrinthine stroke patients remain untreated due to a missed diagnosis; however, data on the rate of missed diagnosis are lacking.

A prognosis in cases with a suspected labyrinthine stroke that was based on MRI-DWI-positive lesions in the vertebrobasilar territory was variable, ranging from no to full recovery in individual patients [14,29,40,44,57,60,61,62,66,67,69,80,83]. In two larger studies on AICA strokes, the authors concluded that the long-term outcome of SSNHL was relatively good [14,29]. Specifically, in one study, 17 out of 21 patients (81%) who were followed for at least 1 year after onset of SSNHL had a partial (n = 10) or complete (n = 7) recovery of hearing, with an average (±1 standard deviation [SD]) improvement of 26 dB in air-conducted and 21 dB in bone-conducted thresholds on pure tone audiometry [14]. In the second study, 63% (39/62) of the patients who were followed for at least 1 year after the onset of SSNHL showed either a partial (n = 24, 62%) or complete (n = 15, 38%) hearing recovery at their last follow-up [29], with an average (±1 SD) improvement from a 66.6 ± 23.6 dB hearing loss initially to a 37.3 ± 17.4 dB hearing loss upon their follow up. Please note that there was a partial overlap in the patients included in these two studies, as the same stroke registry was used with overlapping time periods.

For an MRI-confirmed isolated labyrinthine stroke, information on the outcomes was scarce (being partial only in one case, and overall better for balance than for hearing in the other case [49,74]). In a case of labyrinthine hemorrhage, recovery from SSNHL was reported to be often poor or absent [45,46]. In our review, we observed a variable outcome in cases with SSNHL due to a labyrinthine hemorrhage, ranging from no recovery to full recovery [35,36,42,43,90]. Overall, recovery was reported to be considerably delayed, with no recovery even after 1 month in one study [37]. In single-case studies, recovery of vestibular function was superior compared to hearing, and this was true for both a labyrinthine hemorrhage [46,70,71] and ischemia [60,74].

## 12. Conclusions

Patients who presented with acute vertigo or dizziness in combination with new-onset unilateral hearing loss pose a diagnostic challenge. While the combination of a unilateral hearing impairment and an impaired aVOR ipsilaterally points to a combined cochleovestibular inner ear pathology, central lesions may mimic this pattern. Specifically, ischemic or demyelinating lesions along the central auditory and vestibular pathways at the level of the dorsal/dorsolateral pontomedullary junction and the pons may demonstrate very similar findings. In order to distinguish peripheral from central pathologies and to identify the underlying causes of the symptoms, a combination of structured history taking (asking the patient about focal neurologic symptoms), targeted neuro-otologic examination (including the HINTS+ algorithm), laboratory audiovestibular testing, and a brain MRI is essential. For cases with confirmed central ischemic lesions that do not involve the auditory and vestibular pathways, labyrinthine ischemia is highly likely. Importantly, for confirming a labyrinthine ischemic stroke, a delayed MRI, including a 3D-FLAIR 4 h post contrast sequence, is necessary, as recently demonstrated in single-case reports [49,74,94,95]. The role of delayed MR imaging of the labyrinth, however, should be further investigated in larger prospective studies. Furthermore, access to such advanced imaging protocols may be limited by the resources available, and the economic impact of additional imaging should be considered, as well. But, patients with stroke-mimicks may present with acute audiovestibular symptoms. This is also true for labyrinthitis, vestibular migraine attacks (which may be accompanied by ear symptoms, including aural fullness or subjective hearing impairment), a perilymph or labyrinthine fistula, vestibular schwannoma, as well as (albeit rarely) for acute thiamine deficiency. Furthermore, acute drug intoxication with neuroleptics or antiepileptic drugs may present with acute combined audiovestibular symptoms.

Noteworthy, (transient) audiovestibular symptoms with no accompanying focal neurologic signs and symptoms and with negative MRI-DWI may be warning signs for a pending stroke, thus should lead to an immediate stroke workup and suitable secondary prevention measures, including antiplatelet therapy or anticoagulation, according to the guidelines [110].

## Figures and Tables

**Figure 1 brainsci-15-00725-f001:**
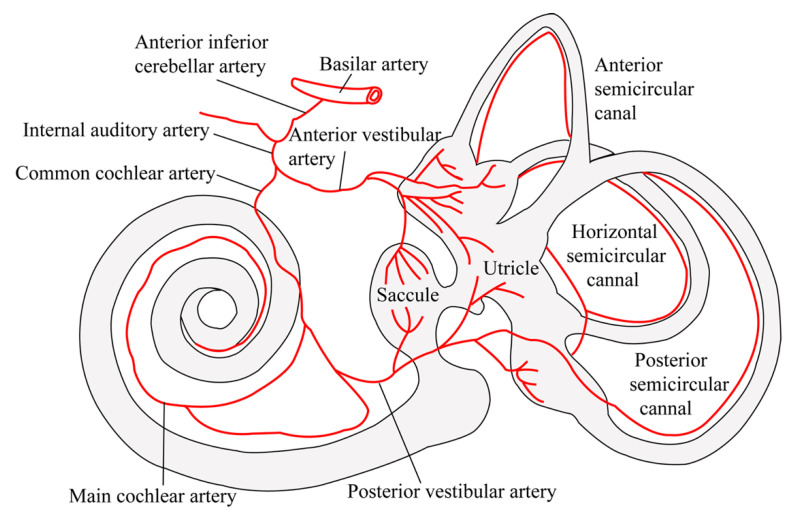
Vascular anatomy of the internal auditory artery, a branch of the anterior inferior cerebellar artery (AICA), and its branches. These supply the inner ear labyrinth, vestibule, and cochlea structures responsible for movement and auditory perception, respectively. This figure was originally published in 2009 by Kim and Lee [18], © Georg Thieme Verlag KG.

**Table 1 brainsci-15-00725-t001:** Key findings from included studies.

	*n* (Studies)	*n* (Patients)
Study population		
Acute cochlear symptoms (SSNHL) [14,29,31,32,33,34,35,36,37,38,39,40,41,42,43,44,45,46,47,48,49,50,51,52,53,54,55]	27	2689
Acute vertigo/dizziness (AVS) [30,56,57,58,59,60,61]	7	86
SSNHL and AVS [62,63,64,65,66,67,68,69,70,71,72,73,74,75,76,77,78,79,80,81,82,83,84,85,86,87,88,89,90,91,92,93,94,95,96,97,98]	37	121
SSNHL or AVS [99]	1	115
All	72	3011
Imaging performed		
MRI 3D-FIESTA [67,93]	2	32
MRI 3D-FLAIR acute [32,35,36,37,38,39,43,50,51,54,69,70,71,79,87]	15	1761
MRI 3D-FLAIR acute + 4 h post contrast [42,49,53,74,94,95]	6	56
MRI-DWI/T1/FLAIR [14,29,30,33,34,40,41,44,45,46,47,48,52,55,56,57,58,59,60,61,62,63,64,65,66,68,72,73,75,76,77,78,80,81,82,83,84,85,86,88,89,90,91,92,96,97,98]	47	907
None [31,99]	2	256
Vestibular symptoms (vertigo, dizziness)		
Yes		1383
No		1094
Not reported		534
Cochlear symptoms (hearing loss)		
Yes		2894
No		115
Not reported		0
Focal neurologic symptoms		
Yes		150
No		2383
Not reported		478
Subtle oculomotor findings		
Yes		163
No		860
Not reported		1988
MRI-based diagnosis		
Peripheral, inner ear disease		
Labyrinthine ischemia confirmed on MRI (3D-FLAIR 4h post contrast [49,74,94,95] or 3D-VISTA post contrast [98]) *		5
Ischemia of the vestibulocochlear nerve [66,96] †		2
Labyrinthine hemorrhage [32,33,34,35,36,37,38,39,41,42,43,45,46,47,48,50,59,64,65,68,70,71,75,78,79,81,85,86,87,88,90,91]		437
Inflammatory inner ear disease [32,34,38,48,50,84]		106
Inner ear trauma [84]		1
Tumor (cerebellopontine angle) ‡		38
Other peripheral disorders §		550
Idiopathic [31,32,34,38,39,43,45,47,48,50,51,53,54,93]		1552
All peripheral cases		2691
Central diseases		
Ischemic stroke		
AICA territory [14,29,30,40,44,56,58,61,66,69,72,76,77,80,84,92,93,96,97]		123
PICA territory [14,98]		7
Multiple territories		
Combined AICA and other territories [14,29,60,62]		60
Combined, no AICA involvement [14,29,30,98]		14
Territory not reported [52,57]		31
Other central disorders ||		88
All central cases		323
Findings in cases with suspected labyrinthine ischemia		
Confirmed ischemic stroke (vertebrobasilar central) [14,29,30,40,44,49,52,56,57,58,60,61,62,66,69,72,76,77,80,84,89,92,93,96,97,98]		235
Audiovestibular symptoms potentially explained by MRI lesions ¶ [14,30,76,80,92,96,97]		34
Prodromal audiovestibular signs or symptoms # [14,52,62,72,77,97]		46
Recovery of audiovestibular symptoms		
No recovery		223
Slight recovery		19
Partial recovery		395
Full recovery		37
Full recovery within 24 h (TIA)		1
Partial recovery of vestibular sx, no recovery of cochlear sx.		5
Not reported		2331

Abbreviations: AICA = anterior inferior cerebellar artery; AVS = acute vestibular syndrome; DWI = diffusion-weighted imaging; FIESTA *=* fast imaging employing steady-state acquisition; FLAIR = fluid-attenuated inversion recovery; PICA = posterior inferior cerebellar artery; SSNHL = sudden sensorineural hearing loss; sx = symptoms; TIA = transient ischemic attack; VISTA = volume isotropic turbo spin-echo acquisition. * No central MRI-DWI lesions in three patients [74,94,95], one DWI-positive lesion in the anterior circulation in one patient [49], and inferior frontal (medial and anterior cerebral artery border zone, left side) and posterior cerebral artery (right side) periventricular white matter lesions in one patient [98]. † In both cases, a combined peripheral (vestibulocochlear nerve) and central (AICA territory) ischemic lesion was detected. These two cases were counted in both the peripheral and central categories, resulting in a total of 3013 diagnoses for 3011 patients. ‡ Tumors identified included vestibular schwannoma (n = 20) [34,47,48,84], meningioma (n = 2) [48], arachnoid cysts displacing the eighth nerve (n = 6) [34,48], and unspecified tumors (n = 10) [38]. § Other peripheral disorders included unspecific signal alterations on MR imaging of the labyrinth (high inner ear protein vs. labyrinthine hemorrhage vs. labyrinthine stroke) (n = 388), decompression sickness (n = 115) [99], neurovascular conflict (n = 22) [48,82], fibrosis of the left posterior semicircular canal 12 days after onset of symptoms (n = 1) [67], fibrotic scare of the vestibulocochlear nerve on histopathology 7 years after acute audiovestibular symptoms and normal MRI-DWI (n = 1) [73], superior semicircular canal dehiscence syndrome (n = 15) [48], endolymphatic hydrops (n = 3) [48], cochlear nerve hypoplasia (n = 1) [48], congenital inner ear deformity (n = 2) [38], vertebrobasilar dolichoectasia (n = 1) [63], and transient labyrinthine ischemia (n = 1) [83]. || Other central disorders included that were not further specified were vascular lesions (stroke, hemorrhage, aneurysm, cerebral vein thrombosis) (n = 14) [48], leukoaraiosis (n = 23) [48], congenital malformations (n = 6) [48], incidentalomas (n = 40) [48], demyelinating lesions in the pons (n = 1) [34] and in the middle cerebellar peduncle (n = 1) [89], and cerebral vein thrombosis (n = 3) [55]. ¶ Remained unclear in two studies with a total of 31 subjects [52,96]. # With either an MRI-DWI that was negative at that time [62,72,77] or no MRI at that time but with delayed focal neurologic symptoms and then confirmation on MRI-DWI [14,52,97].

## Data Availability

The raw data supporting the conclusions of this article will be made available by the authors on request.

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
