# Peer review of "The Challenge of Diagnosing Labyrinthine Stroke—A Critical Review"

_brainsci, 2025, doi:10.3390/brainsci15070725_

Round 1
Reviewer 1 Report
Comments and Suggestions for Authors
Assessment of "The challenge of diagnosing labyrinthine stroke – a critical review":
I enjoyed reading this paper, it is very easy to read. I only have a few points below that should be easy for the authors to address, either directly in the manuscript or in a reply as to why my points are not relevant.
This paper presents a large systematic compilation of scattered litterature on labyrinthine stroke cases, analyzing 3,011 patients from 72 studies.
Robust methodology: Systematic MEDLINE search with clear and reasonable inclusion/exclusion criteria. The author did a great job at with a transparent reporting of study limitations.
Prodromal patterns: The identification of audiovestibular symptoms as warning signs for vertebrobasilar stroke (46 cases identified) provides valuable clinical insight as labyrinthine symptoms can herald vertebrobasilar stroke. It's a point that deserves more attention in the future, in my personal opinion.
Advanced imaging protocols: The emphasis on delayed 3D-FLAIR MRI with contrast for confirming labyrinthine ischemia represents evolving diagnostic approaches, as standard MRI protocols miss labyrinthine ischemia.
Clinically grounded: The paper addresses diagnostic challenges with sensible/practical recommendations. The clinical relevance and potential impact on patient care make it a solid contribution to the literature that could influence practice patterns.
Cons:
If I read it correctly, only 5 cases had definitive MRI confirmation of labyrinthine ischemia using advanced techniques, pointing to mixed populations. But this is part of our evolving diagnostic criteria over time.
Missing Elements:
1. Treatment Outcomes and Acute Management
- Is there any analysis possible of steroid treatment outcomes (standard SSNHL treatment) within this context? Steroid are usually prescribed, and the authors' point is that strokes are often missed. Same comment for hyperbaric oxygen therapy.
- Just like in SSNHL, how often is labyrinthine stroke untreated?
- Missing comparison of outcomes between treated vs. untreated labyrinthine strokes.
- Temporal patterns: I think there should be a statement of symptom evolution timing that might aid diagnosis.
2. Methodological Issues
- (related to above): Publication bias: No assessment of whether negative/unclear cases are underreported.
- Heterogeneity analysis: Limited discussion of why results vary so widely between studies.
3. Clinical Practice Gaps
- Cost-effectiveness: No discussion of economic implications of extensive MRI protocols or that advanced MRI sequences may not be available in many centers
- Follow-up protocols: Missing standardized approach for monitoring patients
4. Patient Selection and Risk Stratification
- Age/comorbidity factors: Limited analysis of which patients are most likely to have vascular causes. A reference to existing litterature might be all that is needed here if it exists.
- Did the authors attempt a subgroup analysis of the data in the table: Insufficient breakdown by age groups, risk factors
5. Emerging Technologies (this is more speculative)
- Possible emerging blood biomarkers?
- Artificial intelligence: No mention of AI-assisted diagnosis or imaging interpretation
- COVID-19 associations: Given timing, surprising omission of pandemic-related vascular complications
Trivial:
- DWI used on page 2 but not defined until page 4
- VISTA is never defined, even though FLAIR and DWI are defined elsewhere.
- Section 10 header: anD
- "labyrinthine infarction is highly suspicious"... I think you mean the opposite! (suspected)
Author Response
I enjoyed reading this paper, it is very easy to read. I only have a few points below that should be easy for the authors to address, either directly in the manuscript or in a reply as to why my points are not relevant.
This paper presents a large systematic compilation of scattered litterature on labyrinthine stroke cases, analyzing 3,011 patients from 72 studies.
Robust methodology: Systematic MEDLINE search with clear and reasonable inclusion / exclusion criteria. The author did a great job at with a transparent reporting of study limitations.
Prodromal patterns: The identification of audiovestibular symptoms as warning signs for vertebrobasilar stroke (46 cases identified) provides valuable clinical insight as labyrinthine symptoms can herald vertebrobasilar stroke. It's a point that deserves more attention in the future, in my personal opinion.
Advanced imaging protocols: The emphasis on delayed 3D-FLAIR MRI with contrast for confirming labyrinthine ischemia represents evolving diagnostic approaches, as standard MRI protocols miss labyrinthine ischemia.
Clinically grounded: The paper addresses diagnostic challenges with sensible/practical recommendations. The clinical relevance and potential impact on patient care make it a solid contribution to the literature that could influence practice patterns.
Cons:
If I read it correctly, only 5 cases had definitive MRI confirmation of labyrinthine ischemia using advanced techniques, pointing to mixed populations. But this is part of our evolving diagnostic criteria over time.
Reply by the authors: Yes, definitive MRI confirmation of labyrinthine stroke cases is still the exception, pointing to one of the major challenges in the field.
Missing Elements:
- Treatment Outcomes and Acute Management
- Is there any analysis possible of steroid treatment outcomes (standard SSNHL treatment) within this context? Steroid are usually prescribed, and the authors' point is that strokes are often missed. Same comment for hyperbaric oxygen therapy.
Reply by the authors: We thank the reviewer for pointing out the issue of unnecessary treatment due to misdiagnosis. As we were focusing on labyrinthine stroke, we were not extracting data on steroid treatment or hyperbaric oxygen therapy and resulting outcome. In those cases that received a diagnosis of (isolated) labyrinthine stroke information on treatment in general and regarding steroids/oxygen specifically was the exception. Thus, we do not have this information available. Retrieving this information would require a modified literature search and separate data extraction but we agree this would be an interesting future project.
- Just like in SSNHL, how often is labyrinthine stroke untreated?
Reply by the authors: This is an important point. However, data on how often labyrinthine stroke is not diagnosed and thus remains untreated are missing. Whenever a labyrinthine stroke is suspected / diagnosed, the focus of treatment is on secondary prevention, including prescription of antiplatelet therapy / anticoagulation (if indicated), lipid lowering drugs, and – if identified – interventions such as carotid endarterectomy or PFO occlusion. In those cases where a labyrinthine stroke is missed, secondary prevention measures will not be undertaken.
We have added the following sentence in the treatment section (section 11):
«Likely a substantial fraction of (isolated) labyrinthine stroke patients remain untreated due to missed diagnosis, however, data on missed diagnosis is lacking.»
- Missing comparison of outcomes between treated vs. untreated labyrinthine strokes.
Reply by the authors: We do agree that such a comparison would be valuable. However, we did not identify any literature reporting on treatment-related outcome in labyrinthine stroke.
- Temporal patterns: I think there should be a statement of symptom evolution timing that might aid diagnosis.
Reply by the authors: We thank the reviewer for pointing out symptom evolution timing as potential aid in diagnosis. We have added the following statement:
«Thus, timing and symptom evolution may have a critical impact on whether the right diagnosis is made or not in such cases with warning signs. Closely following-up these patients is recommended.»
- Methodological Issues
- (related to above): Publication bias: No assessment of whether negative/unclear cases are underreported.
Reply by the authors: Likely there is a publication bias with negative or missed cases being underreported. However, we do not have any information about the extent of such underreporting.
- Heterogeneity analysis: Limited discussion of why results vary so widely between studies.
Reply by the authors: The focus of included studies varied substantially, ranging from describing large cohorts of patients presenting with SSNHL to patients with acute vertebrobasilar stroke. Furthermore, most studies were retrospective, thus a standardized diagnostic workup and analysis of outcome parameters was missing. We now provide a statement on heterogeneity of studies in section 4:
«Study sample size ranged from 1 to 1,300 patients, with most large studies focusing on SSNHL of mixed etiology, pointing to substantial heterogeneity amongst included studies (most likely related to the retrospective nature of most studies included and the varying aims and inclusion criteria).»
- Clinical Practice Gaps
- Cost-effectiveness: No discussion of economic implications of extensive MRI protocols or that advanced MRI sequences may not be available in many centers
Reply by the authors: This is an important point. We have added the following statement to the conclusion section (section 12):
“Furthermore, access to such advanced imaging protocols may be limited by resources available and the economic impact of additional imaging should be considered as well.”
- Follow-up protocols: Missing standardized approach for monitoring patients
Reply by the authors: We do agree that future studies should use a prospective study design including a standardized initial evaluation and follow-up of patients. We have emphasized the need for prospective study protocols in the conclusions section (section 12):
«The role of delayed MR-imaging of the labyrinth, however, should be further investigated in larger prospective studies.»
- Patient Selection and Risk Stratification
- Age/comorbidity factors: Limited analysis of which patients are most likely to have vascular causes. A reference to existing litterature might be all that is needed here if it exists.
Reply by the authors: We thank the reviewer for pointing this out. Probability of cerebrovascular stroke is linked to various risk factors including age, smoking, arterial hypertension, atrial fibrillation, diabetes and others. Risk of stroke increases with age and with the presence of vascular risk factors, as reviewed e.g. by Kleindorfer et al. (Stroke. 2021;52:e364–e467. DOI: 10.1161/STR.0000000000000375). We have indeed included a brief discussion on the impact of vascular risk factors in the context of acute audiovestibular symptoms, providing specific citations as recommended by the Reviewer (at the end of section 9).
- Did the authors attempt a subgroup analysis of the data in the table: Insufficient breakdown by age groups, risk factors
Reply by the authors: We did not systematically perform subgroup analyses because often the level of detail of the data provided was not sufficient or data was lacking completely to do so.
- Emerging Technologies (this is more speculative)
- Possible emerging blood biomarkers?
Reply by the authors: We did not review the literature on possible emerging blood biomarkers. We do agree that this is a topic that should be addressed with more detail in the future.
- Artificial intelligence: No mention of AI-assisted diagnosis or imaging interpretation
Reply by the authors: We did not systematically review the literature included for AI-assisted analysis. Importantly, a substantial number of articles included were published >10 years ago so this would not have been a relevant focus.
- COVID-19 associations: Given timing, surprising omission of pandemic-related vascular complications
Reply by the authors: We did not identify any cases of (isolated) labyrinthine stroke that was reportedly linked to a SARSCoV2 infection. In our literature search we did initially have several citations reporting on vascular complications in patients with SARSCoV2 infection, but there was insufficient detail in these reports to extract information regarding any specific pattern of stroke. The following statement was added to the results section (section 4):
“Noteworthy, no studies reporting on strokes in SARS-CoV-2 patients contained sufficient detail on stroke location to be included in our review.”
Trivial:
- DWI used on page 2 but not defined until page 4
Reply by the authors: We now introduce the abbreviation when first mentioned.
- VISTA is never defined, even though FLAIR and DWI are defined elsewhere.
Reply by the authors: We now introduce the abbreviation when first mentioned.
- Section 10 header: anD
Reply by the authors: Thank you for spotting this typo, which we have now corrected.
- "labyrinthine infarction is highly suspicious"... I think you mean the opposite! (suspected)
Reply by the authors: we thank the reviewer for pointing this out. We have changed suspicious to suspected.
Reviewer 2 Report
Comments and Suggestions for Authors
The review „The challenge of diagnosing labyrinthine stroke – a critical review“ by Tarnutzer et. al. gives a very nice and important contribution to the field of diagnosing stroke in the vestibular system using different methods. The review is very well written, planned, and structured. It provides a literature screen and describes the diagnosis pathways as well as suggestions for improvements. I have only minor points:
- There are a lot of abbreviations used. Please provide a list, which makes it easier for the reader to understand the methods. The abbreviation DWI is in the introduction not introduced. Please do it and check the others ones whether they are.
- The reference in the legend is number 82. This is not comprehensible to me. Please revise.
- The format in 3. “Methodology” is not consistent. Please revise.
Author Response
The review „The challenge of diagnosing labyrinthine stroke – a critical review“ by Tarnutzer et. al. gives a very nice and important contribution to the field of diagnosing stroke in the vestibular system using different methods. The review is very well written, planned, and structured. It provides a literature screen and describes the diagnosis pathways as well as suggestions for improvements. I have only minor points:
- There are a lot of abbreviations used. Please provide a list, which makes it easier for the reader to understand the methods. The abbreviation DWI is in the introduction not introduced. Please do it and check the others ones whether they are.
Reply by the authors: we have checked that all abbreviations are now introduced when first mentioned. Specifically, this was corrected for DWI and for VISTA. Furthermore, a list with abbreviations used has been added at the end of the main manuscript.
- The reference in the legend is number 82. This is not comprehensible to me. Please revise.
Reply by the authors: we have checked the citation number. 82 does refer to the correct citation (Kim and Lee, 2009 Seminars in Neurology). We have added the names of the authors to make it more comprehensible.
«This figure was originally published in 2009 by Kim and Lee [82], © Georg Thieme Verlag KG»
- The format in 3. “Methodology” is not consistent. Please revise.
Reply by the authors: we have reviewed and revised the format in section 3 (methodology) for consistency. It now reads:
“We searched MEDLINE through PUBMED for English-language articles. The search strategy was designed by a clinical investigator with relevant domain expertise in neurology (AAT). We relied on the following strategy and looked for the listed specific components in all articles: (1) acute / new-onset of symptoms, (2) clinical or radiologic signs of ischemic/hemorrhagic stroke, and 3) involvement of the labyrinth and/or the anterior inferior cerebellar artery (AICA). We then selected a series of textual terms to enter in the search system that would refer to the selected criteria (see supplementary material for details on the search strategy). A manual search of the references of eligible articles was also performed. We did not seek to identify research abstracts from meeting proceedings or unpublished studies. Since the submitted work is a critical review, ethical approval was not necessary.
When reviewing all identified citations for eligibility, we used predetermined exclusion criteria and a controlled methodology to select the relevant studies (see supplementary material for details on the search strategy). The review was conducted by a single rater (AAT). Only English-language articles with original data on human subjects with labyrinthine or AICA stroke, which reported on vertigo, dizziness, gait ataxia and/or hearing loss, were included.”
Reviewer 3 Report
Comments and Suggestions for Authors
This manuscript proves to a valuable reference for neurologists and otologists as it provides a detailed and critical review of diagnostic challenges related to labyrinth strokes. The authors focus on vascular anatomy of inner ear and the limitations of imaging techniques. Although clinical bedside assessments such as the HINTS+ algorithm addresses the current challenges in the diagnosis improvements in certain aspects would make the paper stronger.
Major Concerns:
- Critical appraisal of Included studies: The authors should include evaluation of study qualities considering the heterogeneous nature of the numerous studies included. Small case reports might have limited generalizability, and the limitation should be included.
- Diagnostic Pitfalls in Non-stroke Mimics: Discussion on non-stroke conditions like Vestibular migraine that might mimic labyrinthine ischemia should be included to address the clinical overlap.
Minor Concerns:
- There’s inconsistency in terminologies used. A glossary should be included to improve clarity.
- Redundancy in sections discussing HINTS+ approach includes overlapping content which should be consolidated accordingly.
- Post contrast 3D-FLAIR is known to have certain imaging parameters or timing protocols that are missing.
Author Response
This manuscript proves to a valuable reference for neurologists and otologists as it provides a detailed and critical review of diagnostic challenges related to labyrinth strokes. The authors focus on vascular anatomy of inner ear and the limitations of imaging techniques. Although clinical bedside assessments such as the HINTS+ algorithm addresses the current challenges in the diagnosis improvements in certain aspects would make the paper stronger.
Major Concerns:
- Critical appraisal of Included studies: The authors should include evaluation of study qualities considering the heterogeneous nature of the numerous studies included. Small case reports might have limited generalizability, and the limitation should be included.
Reply by the authors: We do agree with the reviewer that the heterogeneity of the studies included is substantial. This is related to the study inclusion criteria and the study design applied (being mostly retrospective in nature and thus lacking a standardized assessment based on a study protocol), but also linked to the study sample size (ranging from single subject studies to large case series of up to 1300 participants). We now emphasize this limitation more and thus have added a statement in section 4:
“Study sample size ranged from 1 to 1,300 patients, with most large studies focusing on SSNHL of mixed etiology, pointing to substantial heterogeneity amongst included studies (most likely related to the retrospective nature of most studies included and the varying aims and inclusion criteria).”
Furthermore, we would like to point out to the following statement in section 12, that points to limitations emerging from single case study evidence:
“Importantly, for confirming labyrinthine ischemic stroke, delayed MRI including 3D-FLAIR 4 h post-contrast is necessary as recently demonstrated in single case reports [29-32]. The role of delayed MR-imaging of the labyrinth, however, should be further investigated in larger prospective studies.»
- Diagnostic Pitfalls in Non-stroke Mimics: Discussion on non-stroke conditions like Vestibular migraine that might mimic labyrinthine ischemia should be included to address the clinical overlap.
Reply by the authors: We included a brief discussion of stroke mimics presenting with combined audiovestibular symptoms in section 12.
With regards to vestibular migraine, mild cochlear symptoms may accompany vestibular symptoms. Likewise – albeit rarely observed - thiamine deficiency can result in both vestibular and cochlear symptoms. Also, labyrinthitis, perilymph or labyrinthine fistula, vestibular schwannoma, and drug intoxication may cause acute audiovestibular symptoms:
“But also stroke-mimics may present with acute audiovestibular symptoms. This is also true for labyrinthitis, vestibular migraine attacks (which may be accompanied by ear symptoms including aural fullness or subjective hearing impairment), perilymph or labyrinthine fistula, vestibular schwannoma, and (albeit rarely) for acute thiamine deficiency. Furthermore, also acute drug intoxication with neuroleptics or antiepileptic drugs may present with acute combined audiovestibular symptoms.”
Minor Concerns:
- There’s inconsistency in terminologies used. A glossary should be included to improve clarity.
Reply by the authors: a list with abbreviations used has been added at the end of the main manuscript.
- Redundancy in sections discussing HINTS+ approach includes overlapping content which should be consolidated accordingly.
Reply by the authors: As suggested by the reviewer, we have revised the section under discussion (section 5) and have rearranged / rephrased sentences to increase flow and consistency.
- Post contrast 3D-FLAIR is known to have certain imaging parameters or timing protocols that are missing.
Reply by the authors: In those four single case reports that used 4h post-contrast application 3D-FLAIR sequences for imaging labyrinthine stroke, no details on imaging parameters or timing protocols were provided. This is a limitation; however, slightly different parameters are expected for different MRI brands also.
Thus, we cannot provide any more detailed information regarding imaging parameters used and would refer to the manufacturer of the MRI in use.